# Comparison of Mechanical and Antibacterial Properties of TiO₂/Ag Ceramics and Ti6Al4V-TiO₂/Ag Composite Materials Using Combined SLM-SPS Techniques

**Ramin Rahmani [1,*], Merilin Rosenberg [2,3], Angela Ivask [3] and Lauri Kollo [1]**

[1] Department of Mechanical and Industrial Engineering, Tallinn University of Technology, Ehitajate Tee 5, 19086 Tallinn, Estonia

[2] Department of Chemistry and Biotechnology, Tallinn University of Technology, Ehitajate Tee 5, 19086 Tallinn, Estonia

[3] Laboratory of Environmental Toxicology, National Institute of Chemical Physics and Biophysics, Akadeemia Tee 23, 12618 Tallinn, Estonia

\* Correspondence: ramin.rahmaniahranjani@ttu.ee; Tel.: +372-5198-8504

**Abstract:** In present work, the combination of spark plasma sintering (SPS) and selective laser melting (SLM) techniques was introduced to produce composite materials where silver-doped titania (TiO₂) ceramics were reinforced with ordered lattice structures of titanium alloy Ti6Al4V. The objective was to create bulk materials with an ordered hierarchical design that were expected to exhibit improved mechanical properties along with an antibacterial effect. The prepared composite materials were evaluated for structural integrity and mechanical properties as well as for antibacterial activity towards *Escherichia coli*. The developed titanium–silver/titania hybrids showed increased damage tolerance and ultimate strength when compared to ceramics without metal reinforcement. However, compared with titania/silver ceramics alone that exhibited significant antibacterial effect, titanium-reinforced ceramics showed significantly reduced antibacterial effect. Thus, to obtain antibacterial materials with increased strength, the composition of metal should either be modified, or covered with antibacterial ceramics. Our results indicated that the used method is a feasible route for adding ceramic reinforcement to 3D printed metal alloys.

**Keywords:** Ti6Al4V lattice structure; Ag-doped TiO₂ anatase; spark plasma sintering; selective laser melting; additive manufacturing; antibacterial and photoactivity applications

## 1. Introduction

Titania (TiO₂) is one of the very often used ceramic materials in medical, antibacterial, paint, varnish, and pigment applications. Two well-known mineral forms of TiO₂ are anatase and rutile that have motivated interests in electrical conductivity and photocatalytic activity fields. Anatase TiO₂ has been shown to exhibit photocatalytic and thus, self-cleaning, activity under ultraviolet illumination. Photodegradation capability of TiO₂, especially under visible light conditions can be even further enhanced by depositing transition metal dopants like silver (Ag) [1,2]. For enhanced antibacterial effect of those materials, Ti-implants have been coated by TiO₂-nanotubes and Ag-nanoparticles in dental and orthopedic applications [3]. The primary interest in TiO₂ ceramics has been related to its use in thin films or as an additive [4]. The implementation of TiO₂ as a bulk ceramic is mainly restricted due to its high brittleness, having fracture toughness in the order of 2–3 MPa·m$^{1/2}$ [5].

It is possible to reinforce ceramic materials with metals, to improve the mechanical properties of the ceramics. Titanium and its alloys are widely used as biomaterials due to their sufficient biocompatibility, light weight, and high mechanical strength [6]. Titanium has excellent physical properties, its corrosion resistance is utterly known for orthopedic, osteology, and dental applications [7] and it possesses the ability to be 3D printed into complex objects. Research studies have displayed that the powder of Ti6Al4V, a material that is finding increasing use in medical applications has a high potential as a starting material for selective laser melting (SLM), a method that can be used to create various 3D structures [8]. Titania that requires relatively low temperatures for full consolidation, may be successfully combined with 3D printed metals. Combining oxide ceramics with metals by common methods is however complicated due to the inherent incompatibility of the interphases. For example, fusing of wear resistant oxide ceramics on a titanium substrate for implants faces distinct challenges in terms of obtaining strong bonding between the ceramic and the metal [9].

Different methods can be applied for consolidating ceramics and composites based on these. When ultrafine- or nano-structures are desired, spark plasma sintering (SPS) is commonly used. Both nanostructured titania and titania based nanocomposites have been produced by SPS [10–12]. Due to very high heating and cooling rate that can preserve the nanostructure at nearly full density, silver-doped titania could be employed for medical applications where antimicrobial properties, chemical inertness, and wear resistance are needed and also for water purification. Combining with titanium could drastically increase the damage tolerance of the ceramic and also provide added application specific functions. For example, a device of combined $TiO_2$/Ag and Cu/CoNiP was shown to effectively perform as magnetically rolling microrobots for water purification [13].

In this study a combination of selective laser melting and spark plasma sintering was introduced for the production of new versatile metal–ceramic hybrid structures to increase damage tolerance of the material. In the process, SLM was used to first 3D print the periodic titanium lattice structure followed by the embedding of titania–silver composite powder to the lattice and hot consolidation by SPS. Prepared materials were analyzed for mechanical properties under a compression test and antibacterial activity against *Escherichia coli* cells.

## 2. Materials and Methods

### 2.1. Materials

Titanium dioxide powders with BET surface area of 150 $m^2$/g (Figure 1a) and flaky silver powder with the purity of 99.95% (Figure 1b) were purchased from ABCR GmbH and used to produce ceramic materials and as the matrix phase in composites. Titanium dioxide was first ultrasonically deagglomerated under isopropanol. Doping with 2.5wt% of silver was performed by using a bottle mixer at 15 RPM for 24 h. Yttria ($Y_2O_3$) stabilized polycrystalline zirconia ($ZrO_2$) was manufactured in TOSOH corporation (TZ-3Y-E, Tokyo, Japan) and was used to produce control ceramic surfaces for antibacterial tests (Figure 1c). Gas atomized Ti6Al4V alloy powders having a particle size in the range from 15 to 45 μm (Figure 1d) were obtained from TLS Technik Spezialpulver GmbH and were used to print metal lattices.

### 2.2. Specimens Preparation

Ceramic materials of $TiO_2$–2.5% Ag, pure $TiO_2$ and $ZrO_2$ (the latter two as controls for the antibacterial test) were produced using a spark plasma sintering (SPS) machine HP D10 from FCT GmbH. Sintering temperature of 750 °C, pressure of 75 MPa and holding time of 30 min were used to compact the composites. To produce metal–ceramic composite materials, first cylindrically shaped Ti6Al4V lattices (Figure 2, top row) were produced using selective laser melting (SLM50 metal additive manufacturing system from Realizer GmbH). The specimens had a diameter of 20 mm and height of 15 mm. The specimens exhibited diamond type porous lattices with unit cell size from 1 to 2 mm. To prepare metal–ceramic composites, porous titanium lattice was placed in a graphite mold and

ceramic powder of $TiO_2$ supplemented with 2.5% Ag was embedded in the lattices (Figure 2, bottom row). SPS at 900 °C, 75 MPa and during 30 min was used to compact the composites (Figure 3). The height of compacted composite lattices was 3–4 mm and depended on the cell size of initial lattices. Lattices with 1 mm cell size were shrunk down to 4 mm, lattices with 1.5 mm cell size shrunk to 3.5 mm and lattices with 2 mm cell unit size shrunk to 3 mm. For the compressive test, identical SPS conditions were applied for 10 mm diameter and 25 mm height lattice structures.

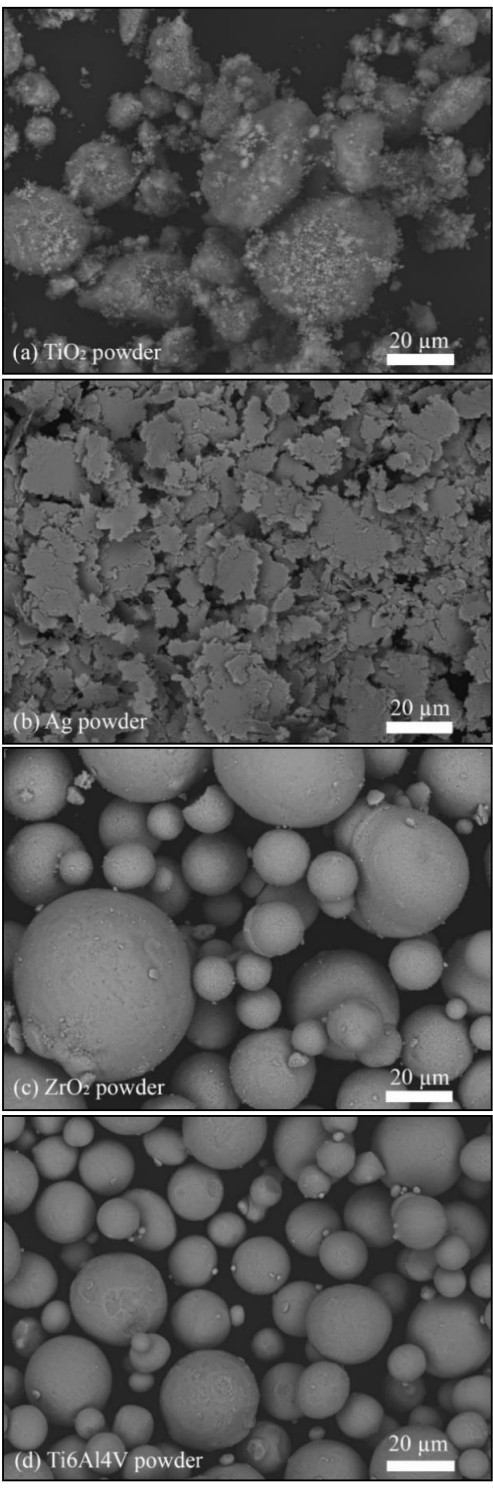

**Figure 1.** SEM micrographs of (**a**) $TiO_2$ anatase, (**b**) Ag, (**c**) $ZrO_2$, and (**d**) Ti6Al4V powders.

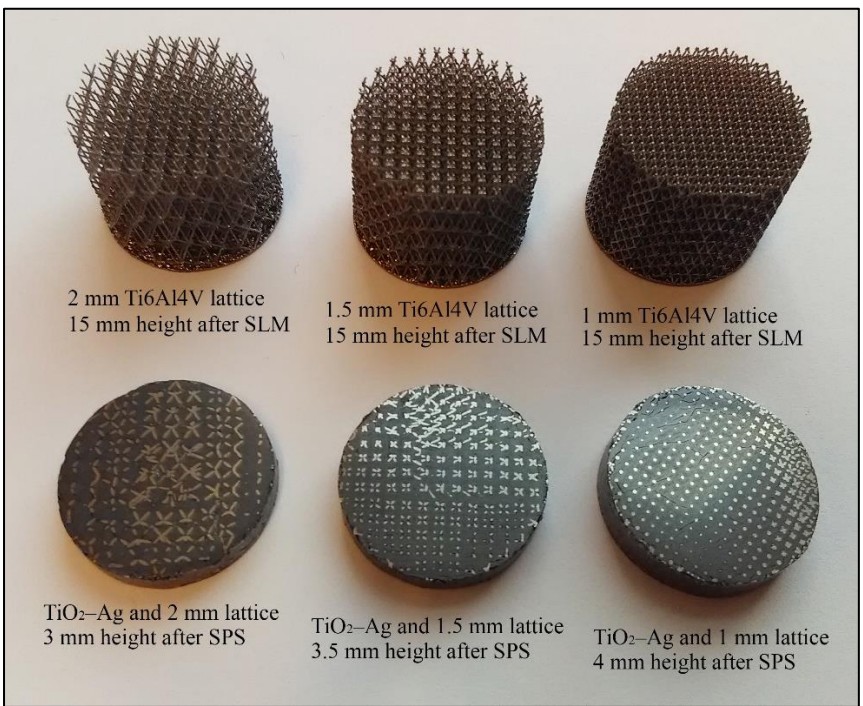

**Figure 2. Top row**: Selective laser melting (SLM) manufactured Ti6Al4V lattice structures with unit cell size and volume fraction of 2 mm and 6%, 1.5 mm and 9%, 1 mm and 16%, respectively (dimensions of lattice structures are 20 mm diameter and 15 mm height); **Bottom row**: Spark plasma sintering (SPS) sintered $TiO_2$–2.5% Ag embedded in the Ti6Al4V lattice structures.

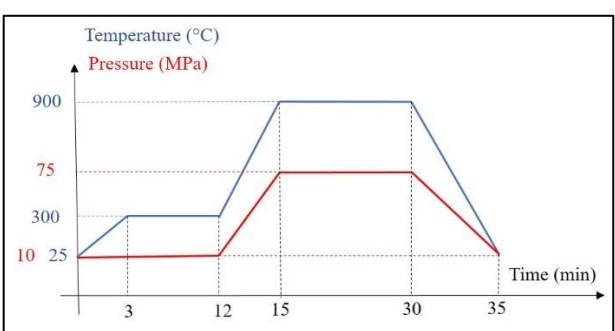

**Figure 3.** SPS conditions for $TiO_2$ powder embedded Ti6Al4V lattice structure.

*2.3. Mechanical and Microstructural Characterization*

Microstructure of the produced materials was examined with a scanning electron microscope (Zeiss EVO MA15, Oberkochen, Germany) equipped with energy dispersive spectroscopy (EDS). Compressive testing of the samples was performed on Instron 8516 servo-hydraulic test machine. The cylindrical samples with diameter of 10 mm and height of 8–9 mm (the height of 3D printed lattice structures was 25 mm which was shrunken to 8–9 mm after SPS) were loaded with a crosshead speed of 0.5 mm/min, according to standard ASTM E9/09.

*2.4. Antibacterial Assay*

A comparative antibacterial assay was carried out for $TiO_2$–2.5% Ag, $TiO_2$, and $ZrO_2$ ceramics as well as for $TiO_2$–2.5% Ag and Ti6Al4V lattice hybrid structures of 1, 1.5, and 2 mm cell sizes (Figure 2). The assay was carried out using an in-house protocol based on ISO 27447:2009 and ISO 22196:2007 standard methods [14] towards a model gram-negative bacterium *Escherichia coli* MG1655. Prior to all experiments, the specimens were sanded and polished, in order to remove potential contaminants and smoothen the surface, then sterilized by autoclaving at 121 °C for 15 min. The material samples were

reused for consecutive experiments. After each test, samples were thoroughly washed with water and 70% ethanol, drained, submerged in 80 mL deionized water and sonicated using Branson Digital Sonifier model 450 (max power 400 W) equipped with horn model 101-135-066R at 25% amplitude for 10 min before autoclaving. *E. coli* culture for inoculum suspension was collected from fresh nutrient agar (5 g/L meat extract, 10 g/L peptone, 5 g/L sodium chloride, 15 g/L agar powder in deionized water) plates incubated overnight at 30 °C, suspended in 500-fold diluted nutrient broth (3 g/L meat extract, 10 g/L peptone, 5 g/L sodium chloride in deionized water) and further diluted with the same medium to optical density of 0.01 at 600 nm. Sterile surface samples were placed on the bottom of sterile 6-well polystyrene plates, inoculated with 50 μL *Escherichia coli* MG1655 suspension and covered with 2 cm × 2 cm × 0.005 cm polyethylene film. Exposure medium was 1:500 diluted nutrient broth. Samples were in parallel either covered by 1.1 mm UVA-transmissive borosilicate glass sheet and exposed to 2–2.5 W/m$^2$ UVA at 315–400 nm spectral range (measured using Delta Ohm UVA probe) effective at the sample level or kept in the dark covered by a 6-well plate lid.

After 30 min and 4 h exposure bacteria were retrieved from samples by repeatedly pipetting 3 mL of neutralizing medium (soybean-casein digest broth with lecithin and polyoxyethylene sorbitan monooleate: 17 g/L casein peptone, 3 g/L soybean peptone, 5 g/L sodium chloride, 2.5 g/L disodium hydrogen phosphate, 2.5 g/L glucose, 1.0 g/L lecithin, 7 g/L nonionic surfactant in deionized water) over the surface, serially diluted in 2 mL volume of physiological saline and from each dilution 3 × 20 μL drop-plated on nutrient agar. Plates were incubated overnight at 30 °C after which colony forming units were counted. The experiment was repeated at least three times for each surface type and time point. Statistical analysis of test results were carried out in GraphPad Prism 7.04 software using one-way ANOVA analysis with Tukey's multiple comparisons test at 0.05 significance level. Due to highly variable and inconsistent results of lattice-embedded samples, these were excluded from statistical analysis at the 4 h time point.

## 3. Results

### 3.1. Structural Study and Mechanical Properties of the Composite Materials and Ceramics

The appearance of TiO$_2$ and TiO$_2$–2.5% Ag ceramic surfaces, and TiO$_2$–2.5% Ag in Ti6Al4V composite materials is shown in Figure 4. The white color of TiO$_2$ ceramic changed to gray when 2.5% Ag was added. However, lighter areas likely with lower Ag content can be seen in TiO$_2$–2.5% Ag ceramic (Figure 4). Under SEM (Figure 5) adequate interphase cohesion with some porosity between ceramic and Ti6Al4V lattice rods was observed. A SEM image (Figure 5a) and digital photograph (Figure 2) show that the 1 mm lattice structure was less bent or distorted subjected to SPS conditions. Critical areas were the interphases between the lattice and ceramic. Although water tightness of the hybrids was achieved, still some porosity at the interphase remained (Figure 5b).

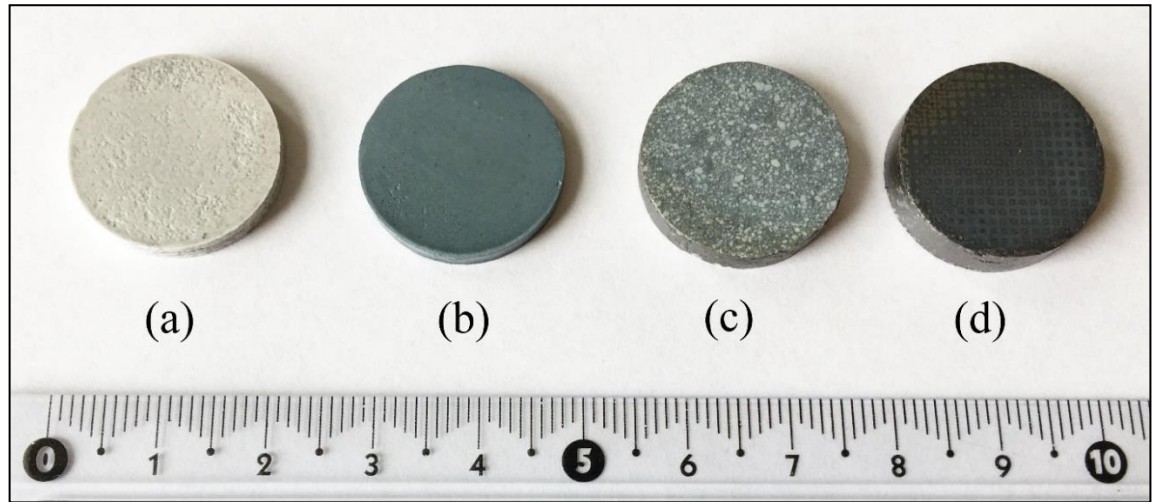

**Figure 4.** Digital photograph of (**a**) $ZrO_2$, (**b**) pure $TiO_2$ anatase, (**c**) $TiO_2$–2.5% Ag, (**d**) composite structure with $TiO_2$–2.5% Ag and Ti6Al4V lattice after sintering (the lattice unit cell size is 1 mm and diameter of samples are 20 mm).

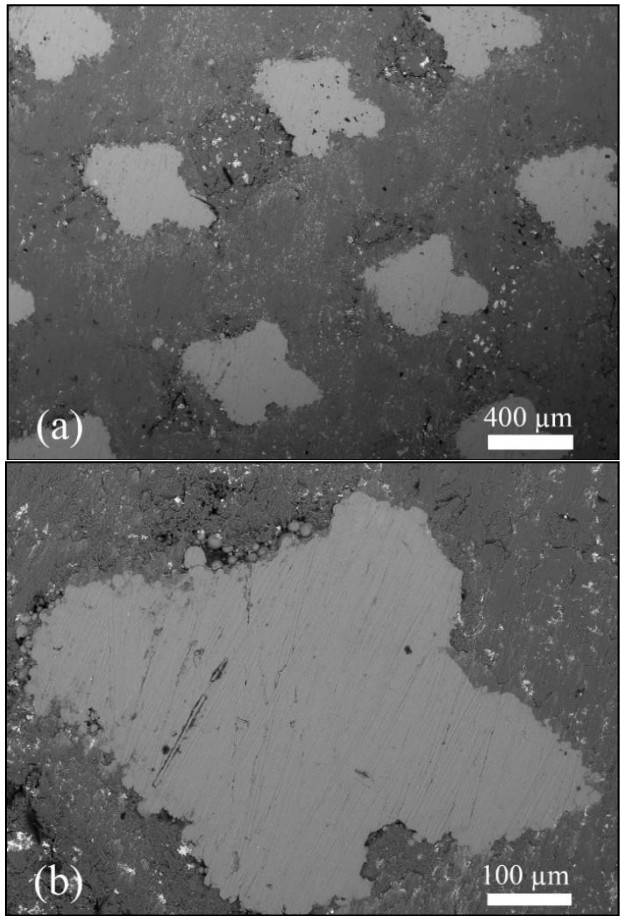

**Figure 5.** SEM micrograph of $TiO_2$–2.5% Ag and Ti6Al4V lattice structure (1 mm unit cell size) taken at ×50 magnification (**a**) and ×200 magnification (**b**).

The EDS elemental mapping results (Figure 6) showed presence of seven elements, namely, Ti 56.45%, O 39.36%, Ag 2.53%, V 0.56%, Al 0.48%, Cl 0.42%, and P 0.20%. The EDS spectrum illustrated acceptable distribution of $TiO_2$ and Ag in composition. Also, rounded Ti6Al4V rods cross-section validated the resistance of 1 mm cell size printed lattice under compression (Figures 2 and 6). To reveal

the damage tolerance characteristics of ceramic–metal composite materials compared to pure ceramics, compression tests were performed (Figure 7). When plain titania ceramic showed brittle fracture, then lattice composite specimens did not catastrophically fail until 25% of deformation. For samples with larger volume fractions of metal phase (1 and 1.5 mm unit cell sizes) in addition to the absence of critical failure, ultimate strength of the composites was significantly higher.

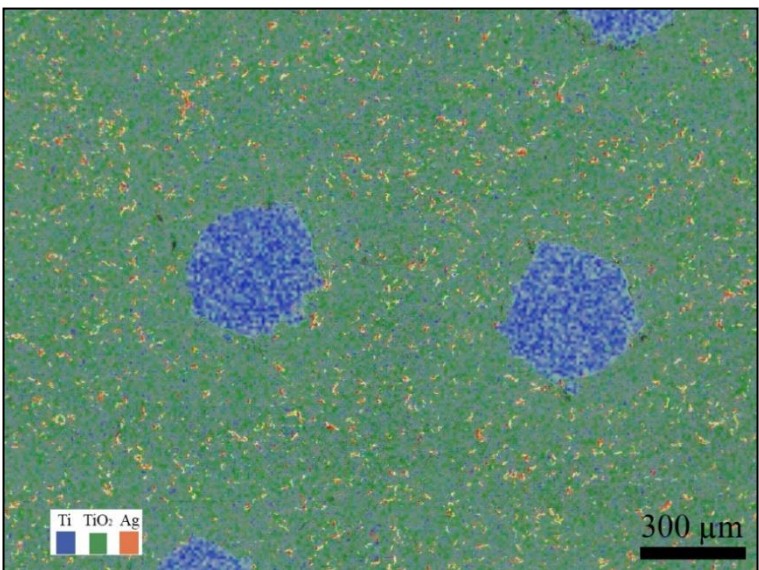

**Figure 6.** Energy dispersive spectroscopy (EDS) color mapping of $TiO_2$–2.5% Ag embedded 1 mm cell size Ti6Al4V lattice structure.

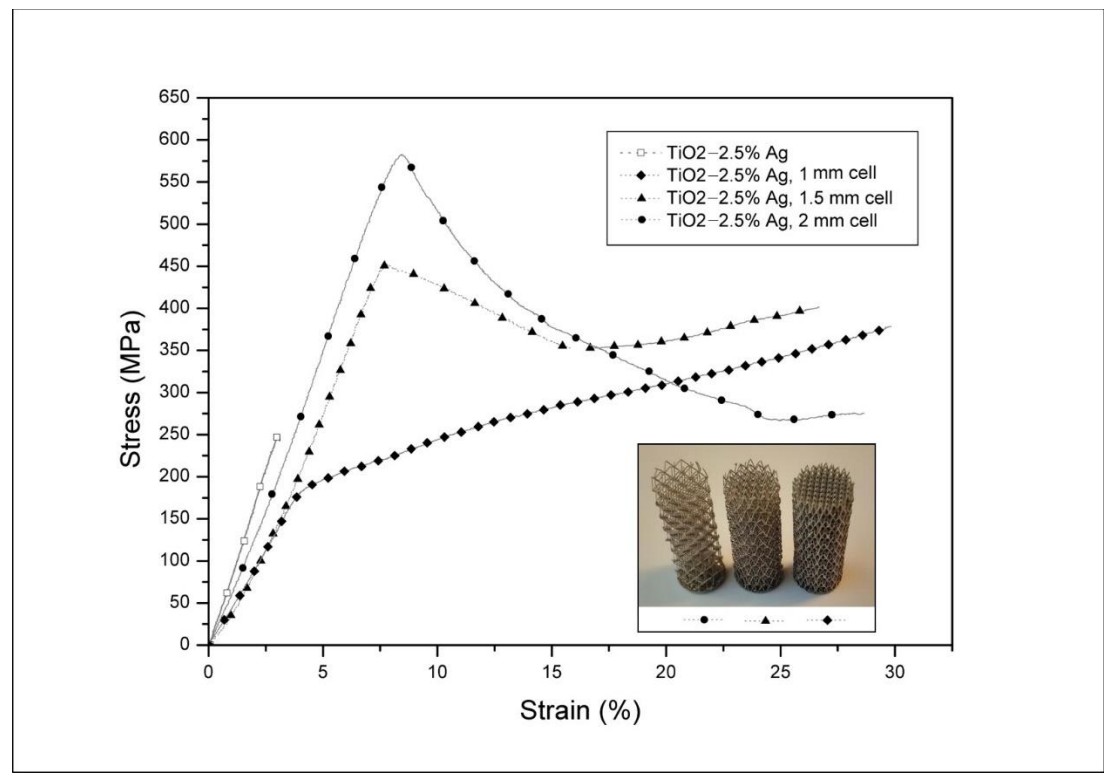

**Figure 7.** Compressive test results for $TiO_2$–Ag without and with different unit cell sizes of lattice structure. Height of lattices were 25 mm and diameter was 10 mm before SPS.

### 3.2. Antibacterial Activity of the Surfaces

Antibacterial activity of the composite and ceramic materials towards *E. coli* MG1655 was evaluated after 30 min and 4 h exposure (Figure 8) while using ceramic zirconia surface as a negative control. Among the tested materials, the highest antibacterial effect in dark conditions was observed for $TiO_2$–2.5% Ag ceramics where >3 logs reduction in *E. coli* viability was observed already within 30 min compared to control ($p < 0.01$) and no viable bacteria detected at the detection limit of about 450 colony forming units (CFU) per surface. As ceramic $TiO_2$ without added silver had no antibacterial effect in dark conditions ($p > 0.05$), we suggest that the effect seen for Ag-supplemented $TiO_2$ ceramics was due to Ag ions released from the material. Results for lattice-embedded $TiO_2$–2.5% Ag surfaces were not statistically different from control after 30 min ($p > 0.05$) and had too high variability to compare results with full ceramic materials after the 4 h time point. The fact that significantly less antibacterial effect was seen for lattice-embedded samples suggests that the release of Ag from those materials was much lower than from ceramic samples.

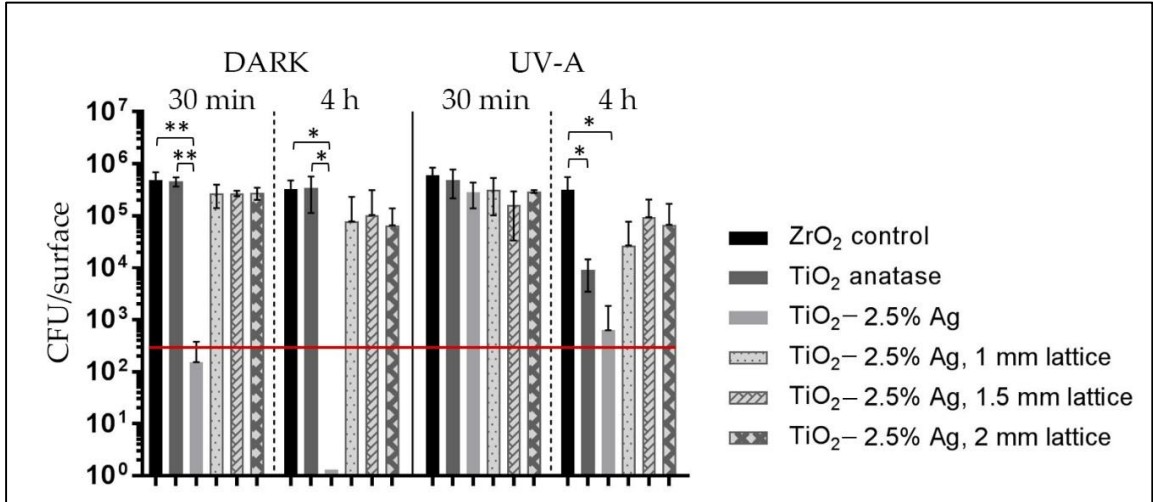

**Figure 8.** Viability of *Escherichia coli* MG1655 on ceramic and composite hybrid surfaces after 30 min and 4 h exposure in the dark and UV-A-illumination. Columns represent recovered viable bacteria as colony forming units (CFU). Mean and standard deviation of at least three independent values is shown on a logarithmic scale and only statistically significant differences ($p < 0.05$) marked on the graph (* $p < 0.05$ and ** $p < 0.01$). Only the upper error bar is shown for samples with >100% SD. Lattice samples are excluded from statistical analysis at 4 h time points due to very high variability. Limit of detection at 458 CFU/surface marked in red.

Due to the photocatalytic nature of $TiO_2$, the samples were assumed to exhibit UV-induced antibacterial effects. Indeed, 4 h exposure of bacteria to the ceramic $TiO_2$ surface under UV-A decreased bacterial viability by 1.6 logs ($p < 0.05$) compared to the control surface. The efficacy of Ag-supplemented ceramic $TiO_2$ surface under UV-A was higher than that of ceramic $TiO_2$ surface but significantly lower than the efficacy of $TiO_2$–2.5% Ag surface in dark conditions. This is likely because UV exposure has been shown to significantly decrease Ag solubility and subsequently, antibacterial activity as has been previously shown for photo-inducible Ag complemented ZnO surfaces [14].

Metal reinforced $TiO_2$–2.5% Ag ceramics showed significantly lower antibacterial effect than what was seen for ceramic surfaces. After 30 min under UV-A, $TiO_2$–2.5% Ag composite surfaces did not exhibit significant antibacterial effects compared with $ZrO_2$ control. After 4 h UV-A exposure, the composite surfaces yielded results that had too high variability to statistically compare them with control surfaces or full ceramic materials. However, according to the general picture, the composite surfaces exhibited slight antibacterial effect as compared to $ZrO_2$ control. These results showed that

while including titanium lattice to TiO$_2$–2.5% Ag ceramic material increased the damage tolerance of the material, it significantly decreased the antibacterial effect of the ceramic material.

## 4. Discussion

Bonding of brittle ceramics to structural metals in assemblies has often remained a challenge, requiring designing for bolting or specific soldering alloys. The results described in this work represent a new approach for bonding, using additively manufactured lattice structure in the interphase of metal and ceramic. The composite structure where TiO$_2$–2.5% Ag was bonded with titanium showed not only increased damage tolerance but also increased ultimate compressive strength when compared to unreinforced ceramics. To explain the increased strength and damage tolerance, ceramic and metal–ceramic samples were subjected to a compressive test (Figure 7). The results showed large fractured pieces in the case of ceramics (Figure 9a), whereas ceramics in the composite sample was fractured into sub-micrometric particles (Figure 9b). For highly brittle material such as TiO$_2$–Ag, a larger content of energy is absorbed in crack initiation for hybrid composites.

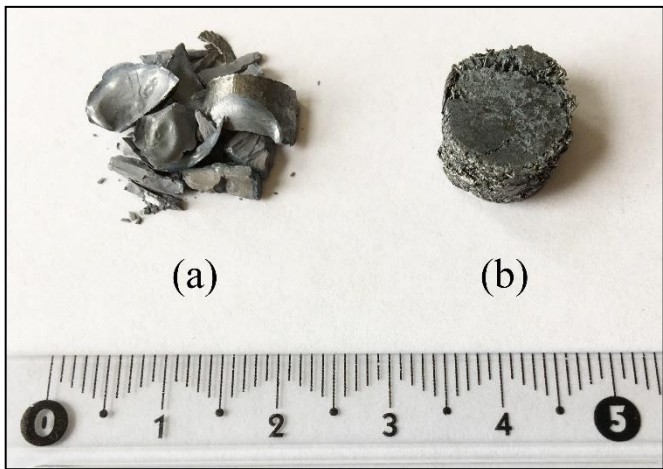

**Figure 9.** Appearance of (**a**) TiO$_2$, (**b**) composite structure with TiO$_2$–Ti6Al4V hybrid after compressive testing (sample diameter was 10 mm and unit cell size was 1 mm).

The proposed failure mechanism of composite hybrids is visualized in Figure 10. Three modes of deformation could be distinguished. During the first mode, at applied strain up to 2–3 percent, energy was absorbed in rearrangement and elastic deformation of the metallic lattice. The elastic modulus of the composite hybrid ($\approx$50–80 GPa) was significantly lower than the elastic modulus of both separated constituents, Ti6Al4V ($\approx$110–120 GPa) and TiO$_2$ ($\approx$230–280 GPa). In the middle region of the elastic part of the compressive loading curve a cracking sound was observed. This was due to the removal of fractures of ceramic elements exposed to the surface (schematically shown in Figure 10, Mode 2). Until the yielding point, surface exposed ceramic elements were gradually removed, and this remained as the main deformation mechanism during plastic deformation of the hybrid, during Mode 3. Additionally, the interior ceramic elements were fractured as shown in Figure 10, Mode 3. The opposing force from metallic lattice (indicated by black arrows in Figure 10) induced back-pressure and fractured ceramic pieces got embedded into a ductile metal lattice. This interaction increased the damage tolerance of the composite hybrid under compressive loading.

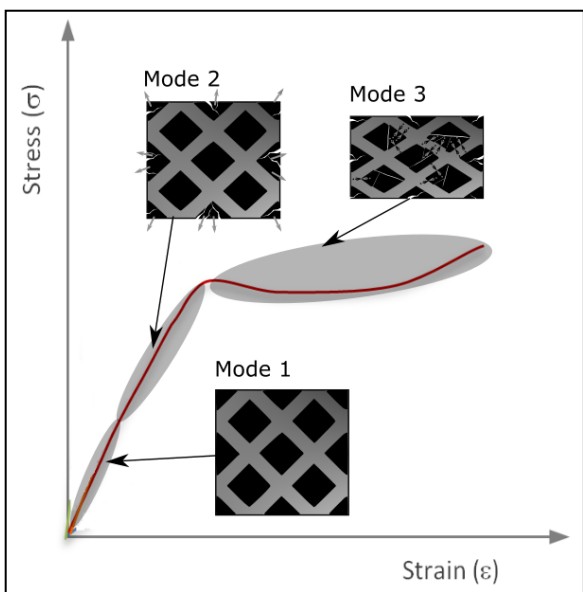

**Figure 10.** Schematic showing fracture mechanisms of metal–ceramic hybrids under compressive loading. Mode 1: rearrangement and elastic deformation of metal lattice; Mode 2: fracturing of ceramic surface elements; Mode 3: fracturing of the interior ceramic elements and embedding of these in ductile metal lattice.

Consequently, three modes of failure of the hybrid composite were differentiated during compressive loading. The increased damage tolerance and compressive strength were attributed to higher input energy needed to fracture the ceramics and interaction between fractured ceramic pieces and ductile metal lattice inside the material. As it was seen on the hybrid specimens after compressive testing (Figure 9b), the ceramic material was removed preferentially at the perimeter of the cylindrical sample. Further strengthening could be achieved if hybrid composite would be surrounded by an additional metallic layer so that ceramic would not be exposed on the outer surface.

Metal 3D printing enables the production of lattice structure objects with different shapes and internal mesostructures. The composite, therefore, can be designed according to existing mechanical loads. Functionally grading in different directions, and integrating solid printed metals with ceramic–metal hybrids could be realized. Furthermore, a metal lattice could be designed so that it acts as a heating element when an electric current is directed through the material. The heating would further enhance the antimicrobial effect of the ceramic.

The compressive test result of $TiO_2$ ceramic and $Ti6Al4V$–$TiO_2$ lattice composite was captured (Figure 10) and showed the benefits of lattice structure perfectly. Finite element analysis prepares an estimation of metal–ceramic composite materials strength produced by combining SLM-SPS technique subjected impact, abrasion or compression loading [15–17]. Numerical simulation illustrated that fracture will occur at around 500 MPa for pure ceramic, while buckling for metallic lattice will start from 1200 MPa (Figure 11). Damage tolerance of $Ti6Al4V$–$TiO_2$ composite depends on the densification of ceramic (SLM parameters) and metal–ceramic bonding phase (SPS parameters).

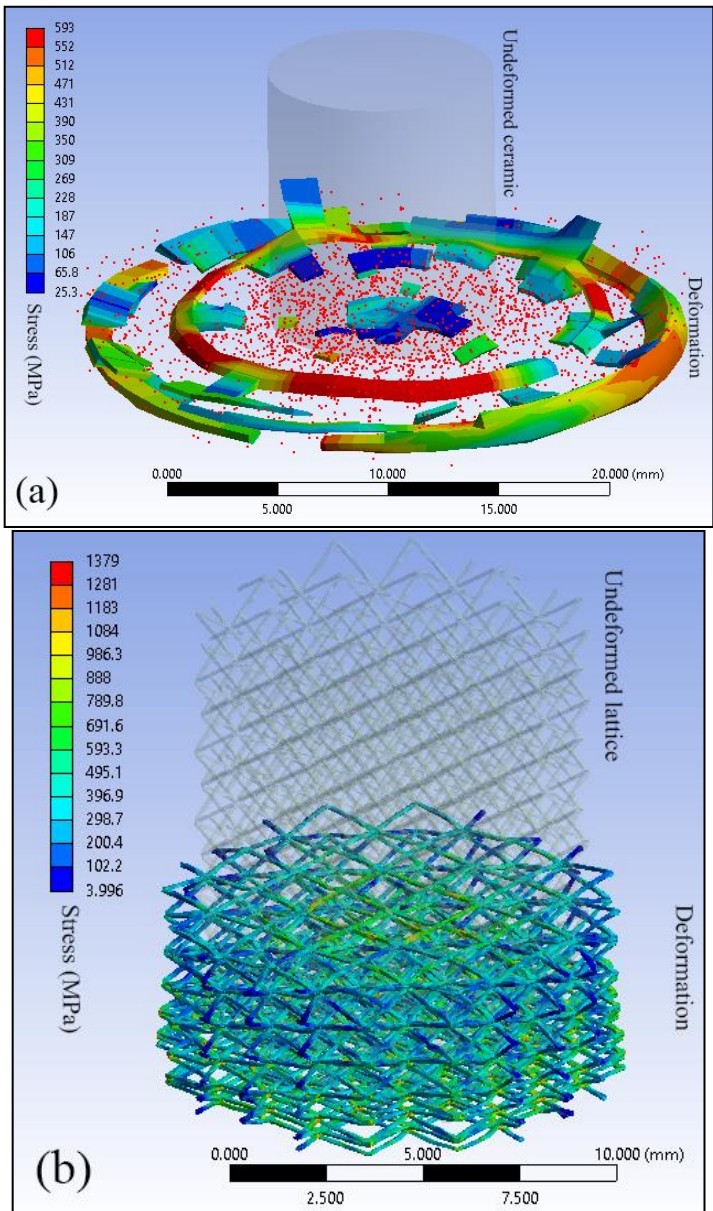

**Figure 11.** Compressive strength modelling of (**a**) TiO$_2$ ceramic, (**b**) Ti6Al4V lattice structure. Simulation conditions and dimensions are identical for the ceramic and lattice structures.

The present work used SPS as a consolidation method, which can produce only simple, cylindrical shapes. Using other hot consolidation methods as hot isostatic pressing or hot forging, 3D shaped hybrid composites could also be manufactured. The ability to produce composites with complex shape could provide new solutions for a number of applications in the field of metal–ceramic hybrids.

This study was unique as there are no prior studies reporting on antibacterial effects of hybrid composite metal–ceramic materials. However, studies on antibacterial effects of Ag-containing ceramic materials have been previously published [18]. In general, those studies have shown the relationship between the amount of silver in the ceramic surface and antibacterial activity [19] and the importance of segregation and agglomeration of silver on the surface for improving antibacterial efficacy [20–22]. Similar observation was also done in this study but only in dark conditions. Under UV-A, the effect of added Ag to ceramics had significantly smaller effect than in dark conditions. We suggest that this was due to Ag ions which drive the antibacterial effect of Ag–ceramic surfaces [23–25] being reduced back to elemental Ag onto the surfaces [14]. Compared with TiO$_2$–Ag ceramic material, the

antimicrobial effect of hybrid surfaces was drastically reduced (Figure 8). This change could not be only explained by reduced area of antimicrobial $TiO_2$–Ag surface in composite material as titanium metal lattice occupied 15% to 25% of the surface, depending on the sample. In almost all cases, both in dark and under ultraviolet exposure the antimicrobial effect of composite surfaces was orders of magnitude lower when compared to the fully ceramic surface. The reason for this was not clear, but it could be assumed that there was a combination of direct surface contact and soluble silver toxicity in effect, both dependent on silver exposure at the material surface. These results indicated that if the hybrid composite needed to be exposed on the surface, the metal composition would also need to be chemically modified for an antimicrobial effect. Otherwise, we suggest that in order to preserve the antibacterial activity of the composite material and reduce variability in antibacterial activity results, a thin layer of pure ceramic material should be added to the surface of the composite.

## 5. Conclusions

An approach to produce titanium/silver-doped titania composites was introduced, by combining of SLM and SPS techniques. The metallic lattice structures were 3D printed, embedded with $TiO_2$–Ag ceramic powder and consolidated by SPS. Compression strength and damage tolerance of the composites were shown to increase significantly when compared to $TiO_2$–Ag ceramics. No collapsing of the composites was seen at up to 25% of deformation in the compressive test. Adding metallic lattice to ceramic silver-doped titania material however decreased the antibacterial effect compared with ceramics only, significantly. Thus, we suggest that the composition of metal that is used to produce the lattice should be chemically modified or a thin layer of pure ceramic material should be added to the surface of the composite.

**Author Contributions:** "conceptualization, R.R. and M.R.; methodology, L.K. and A.I.; software, R.R.; validation, M.R., A.I. and L.K.; formal analysis, R.R.; investigation, R.R. and M.R.; resources, L.K. and A.I.; data curation, M.R.; writing—original draft preparation, R.R.; writing—review and editing, R.R., M.R., A.I. and L.K.; visualization, A.I.; supervision, L.K.; project administration, A.I.; funding acquisition, L.K.".

**Funding:** This research was funded by the Estonian Ministry of Education and Research (IUT 19-29; PUT 748; IUT 23-5; base funding provided to R&D institutions B56 and SS427; M-ERA.NET DURACER project ETAG18012); The European Regional Fund, project number 2014-2020.4.01.16-0183 (Smart Industry Centre) and ERDF project TK134.

**Acknowledgments:** The authors would like to thank Mart Viljus for the help with EDS mapping.

**Conflicts of Interest:** The authors declare that there are no conflicts of interest.

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
