# Peer review of "Comparison of Mechanical and Antibacterial Properties of TiO2/Ag Ceramics and Ti6Al4V-TiO2/Ag Composite Materials Using Combined SLM-SPS Techniques"

_metals, doi:10.3390/met9080874_

Round 1

Reviewer 1 Report

There are several issues that should be addressed in the manuscript before further consideration for publication.

Suggest to describe the SLM process in more detail, using ASTM/ISO terminology

- Lee et al. (2018), 3D bioprinting processes: A perspective on classification and terminology, International Journal of Bioprinting 4(2),151

While SLM is a big part of the manuscript, there is lack of discussion and description of what is done with this process, especially for Ti6Al4V which is already a very established material. For example, what process parameters are used?

- Khorasani et al. (2019), A comprehensive study on variability of relative density in selective laser melting of Ti-6Al-4V, Virtual and Physical Prototyping  14(4), 349-259

There are numerous  method that have been used to produce composite using SLM. Why is SPS chosen? How are the other methods in comparison?

- Wu et al. (2019), Effects of TiO2 doping on microstructure and properties of directed laser deposition alumina/aluminum titanate composites, Virtual and Physical Prototyping 14(4), 371-381

- Yu et al. (2019), Particle-Reinforced Metal Matrix Nanocomposites Fabricated by Selective Laser Melting: A State of the Art Review, Progress in Materials Science 104, 330-379

- Sing et al. (2017), Fabrication of titanium based biphasic scaffold using selective laser melting and collagen immersion, International Journal of Bioprinting 3(1), 65-71

How are the design of the lattices chosen? 

Any benchmarking done to other work on composite lattices?

Reviewer 2 Report

In this work, the authors demonstrate Ag doped TiO2 reinforced with Ti6Al4V for 3d printed metal alloy. The authors evaluated the effects of the reinforcement on the mechanical strength, and anti-bacterial effects, demonstrating that a compromise should be used to achieve both mechanical strength and anti-bacterial properties simultaneously. The paper is well-written and interesting for the readers of this journal.

Some comments:

1. The digital photographs should also have scale bars. The top subfigure in Figure 1 does not have an actual bar to represent the 20 microns.

2. Figure 11 is used in the discussion section but appears in the conclusion. Please move it to the correct place where it is referenced.

3. UVA is used without description.

4. Caption for Fig.5 should explicitly describe each subfigure (a) and (b).

Reviewer 3 Report

The paper reports the mechanical and antibacterical properties of Ti6Al4V-TiO2/Ag composites obtained by SLM-SPS, as compared to those of TiO2/Ag ceramics.

Points of strength: interesting and fairly new approach to the formation of metal-ceramic composites using SLM and SPS.

Points of weakness: the results should be rearranged (for example, the order and the number of figures should be the same of their citation in the text), and they might be improved.

Introduction

1. Line 40: “orthotropic”: I suppose you mean “orthopedic”

2. Line 57-58: “Due to…” the sentence has not a subject. I suppose that the full stop is a mistake.

Materials and methods

3. Line 76: “friction”: I suppose you mean “fraction”

4. Figure 2: In this figure a sample with 0.75 mm unit cell size (letter d) is shown, but the results obtained with this sample type were not discussed.

4. Regarding specimen preparation, please, specify in section 2.2 the sizes of all the SLM prepared specimens (for microstructure analysis, compressive tests and antibacterial assay), and the final height of the different compacted specimens. I suggest you to show Fig 8a in this section, together with Fig. 2, in order to depict the different obtained structures.

5. Line 99: it is reported that “ceramic powders (TiO2 or TiO2 supplemented with 2.5% Ag) were embedded in the lattices.” No data of titanium lattice + TiO2 only were reported in the paper.

6. Lines 110-111: the characteristics of the lattice structure of the specimens for the compressive tests should be reported in section 2.2.

7. How many compressive tests did you perform to assess the results?

8. Regarding antibacterial assay, it is not reported whether the surfaces of the specimens were ground or polished after extracting them from the graphite mold and before autoclaving them. A slight polishing of the surface is useful to remove eventual contaminants due to the SPS process, which can influence antibacterial results.

Results

9. Did you assess whether an anatase to rutile transformation occurred after SPS? A X-ray diffraction analysis of the surface of the samples for assessing phase composition might be performed.

10. The order and the number of figures should be the same of their citation in the text. Fig. 7 should be Fig. 6.

11. Did you find Cl and P also in TiO2 powders?

12. I suggest you to report here the surface fractions of titanium and TiO2+Ag as a function of different lattice structure. This is an important datum, since it is only TiO2+Ag the active surface for the antibacterial effect, and it is only briefly reported in the discussion (lines 277-278).

13. Lines 166-168: it is reported that “For samples with larger volume fractions of metal phase (1 and 1.5 mm unit cell sizes) in addition to the absence of critical failure, also yield strength of the composites was significantly higher.” As a matter of fact, the sample with the highest yield strength is that with 2 mm cells. Perhaps, you referred to the ultimate strength, and not to yield strength.

14. The characteristics of the fractured specimens at the end of compressive tests and Figure 9 should be analysed here and not in section 4.

15. Regarding Fig. 8 and its caption, please, use 4 h also in the figure, not 240 min.

16. Line 210: it is reported “Limit of detection at 458 CFU/surface marked in red”. No red mark is present in the figure.

Discussion

17. Line 268: it is claimed that “numerous studies on antibacterial effects of Ag-containing ceramic materials have been published”, but only one reference is present. If you refer to references within [18], please, specify it.

English should be carefully checked.

Round 2

Reviewer 1 Report

NA